



# CAR36, a regional high-resolution ocean forecasting system for improving drift and beaching of Sargassum in the Caribbean Archipelago.

Sylvain Cailleau[1], Laurent Bessières[1], Léonel Chiendje[2], Flavie Dubost[1], Guillaume Reffray[1], Jean-Michel Lellouche[1], Simon van Gennip[1], Charly Régnier[1], Marie Drevillon[1], Marc Tressol[1], Matthieu Clavier[1], Julien Temple-Boyer[1], Léo Berline[3].

[1]Department Operational Oceanography, Mercator Ocean international, Toulouse, 31400, France
[2]Master II In Oceanography and Applications, University of Abomey-Calavi, Cotonou, Benin
[3]Aix Marseille Univ, Université de Toulon, CNRS, IRD, MIO, Marseille, France

*Correspondence to*: Sylvain Cailleau (sylvain.cailleau@mercator-ocean.fr)

## Abstract

The stranding of sargassum seaweed on the Caribbean Archipelago beaches constitutes a real socio-economic, ecological and health problem. Météo-France currently operates a model of sargassum drift forecasts (called MOTHY) forced by ocean currents from the global analysis and forecasting system GLO12 at 1/12° (~ 9 km over the Caribbean) operated by Mercator Ocean International (MOi). In order to improve the Météo-France drift forecast, MOi has developed a regional high-resolution ocean forecasting system CAR36 at 1/36° (~ 3 km) centred on the Caribbean Archipelago region. In addition to a finer spatial resolution, this system was designed to resolve some higher frequency signals such as the tidal forcing and to use hourly atmospheric forcing including the inverse barometer effect.

Here the added value of the CAR36 system relative to GLO12 is evaluated with particular focus on the reproduction of meso- and sub-mesoscale structures representing key features of the Caribbean region dynamics and therefore sargassum transport. The realism of the local dynamics was examined with standard statistical validation diagnostics using satellite data (Sea Surface Height, Sea Surface Temperature, Sargassum detection) and drifting buoys, together with more process-oriented diagnostics such as eddy detection and tracking across the domain.

GLO12 and CAR36 hindcast simulations were compared over the year 2019. CAR36 showed marginally better scores using OceanPredict diagnostics (https://oceanpredict.org/). The dynamics of a westward-propagating North Brazil Current (NBC) eddy from its entry into the domain to its dissipation was found to be more realistic in CAR36, especially at the end of its lifetime when it collides with the Caribbean Archipelago. The transfer of kinetic energy from the eddy dissipating westward into the Caribbean Sea was diagnosed as more pronounced for CAR36 corresponding to filamentary structures crossing the Caribbean Archipelago and resulting in part from the friction of the NBC eddy along the islands to the east. Using detection from satellite, aggregation of Sargassum around eddies or along filaments suggests that CAR36 should be able to improve the algae drift forecasts.



## 1. Introduction

In recent years, the exponential growth and beaching of Sargassum – a pelagic algae that strands in large quantities on beaches
– has become an issue of ecological, and socio-economic concern for many islands of the Caribbean region.
Originally concentrated in the so-called Sargasso Sea located in the north-western part of the tropical Atlantic and north of the Caribbean Archipelago, at the convergence zone of the subtropical gyre, large quantities of Sargassum have been observed since 2010 more to the south, along a band stretching from the West African coast to the Gulf of Mexico and referred to as the "Great Atlantic Sargassum Belt" (Wang et al., 2019).
The cause of such recent expansion is not certain but one hypothesis have been put forward recently. An extreme westward wind event in 2010 would have favoured a large transport of Sargassum from the Sargasso Sea to the Eastern Atlantic, which then ended in the equatorial area (Johns et al 2020). Since then, several physical and biogeochemical processes act to produce interannual variability. Nevertheless, the biophysical processes impacting the proliferation of Sargassum and transport to the Caribbean are still not well understood. Few studies have been carried out on the physiology of these species and their response
to the environment. Moreover, waves, wind and currents directly affect the drift of Sargassum rafts (Podlejski et al 2023): the aggregation processes  their extent, shape and thickness (up to 10 m) also play a role in their drift. Improving our knowledge of these different aspects, and taking them into account in numerical models, will allow to better predict algal drift.

The overabundance and stranding of seaweed, particularly in the last decade on the coasts of the Caribbean, Central America
and Brazil, is a real scourge causing major socio-economic, ecological and health problems: the seaweed invades the most popular beaches, harbours and bays, it prevents navigation and local fishing, and once stranded, it gives off a toxic, nauseating gas as it decomposes. This issue is of concern to international and national bodies. The Intergovernmental Oceanographic Commission of the United Nations Educational, Scientific and Cultural Organization (IOC-UNESCO) for the Caribbean and Surrounding Regions (IOCARIBE), its regional alliance with the Global Ocean Observing System (IOCARIBE-GOOS), and
GEO Blue Planet are working with partner organisations and stakeholders to develop a multi-hazard information and warning system focusing in particular on Sargassum drift ( https://sargassumhub.org/ )
At the French level, regarding the mid-term and large-scale forecasting of Sargassum concentration, the FOREcasting seasonal Sargassum Event in Atlantic (FORESEA) project ( https://sargassum-foresea.cnrs.fr/ )  provides monthly forecasts' map over the equatorial Atlantic ocean. Regarding the short-term and fine-scale forecasting, Mercator Ocean international (MOi) has
committed to providing Météo-France with reliable and accurate sub-surface current data over the Caribbean Archipelago region, in order to improve their Sargassum drift forecast and as such better anticipate massive stranding episodes on the islands of Guadeloupe and Martinique. As for any surface drifting objects, (Vic et al 2022) meso- and sub-mesoscale activity likely plays an important role in the aggregation process and the drift of the Sargassum as it has been highlighted by Zhong et al. (2012) in the neighboring area of the Gulf of Mexico. Yet the present Météo-France Sargassum drift model (MOTHY
model, Daniel, 1996; Daniel et al., 2002) underlying hydrodynamic system GLO12 (the Copernicus Marine Service global





ocean analysis and forecasting system at 1/12°; Lellouche et al., 2018; CMEMS product DOI: https://doi.org/10.48670/moi-00016) is of too low resolution to reproduce such structures, stressing the need for a higher resolution alternative. A regional forecasting system centred on the Caribbean Archipelago region (hereafter called CAR36) has therefore been developed at MOi and produces ocean currents at 1/36° horizontal resolution (i.e. approximately 3 km in the area of interest) in real time.

These ocean currents will replace GLO12 ones ) in the Météo-France drift forecast system.

The hindcasts' solution of a reference simulation of CAR36 over the year 2019 is confronted with the hindcasts' solution obtained by GLO12 over the same period and validated by satellite and in-situ observations of sea surface temperature (SST), sea surface height (SSH) and Lagrangian trajectories from the Copernicus Marine Service portal (https://marine.copernicus.eu/fr). Sargassum detection data (Berline et Descloitres, 2021) are also used to estimate the quality

of the simulated currents and to discern areas of algal aggregation.

The paper is organized as follows. The ocean dynamics of the considered area is described in section 2, the main characteristics of the GLO12 and CAR36 systems are presented in Section 3. The validation methodology and observation data used for the models' assessment are presented in section 4. Results of scientific evaluation of CAR36 and GLO12 systems, including comparisons with observations, are given in Section 5. Sections 6 and 7 contain final discussion, conclusion and propsects.

**2.    Ocean dynamics around the Caribbean**

The ocean circulation of the northeast Caribbean region is dominated by westward currents corresponding to a branch of the subtropical gyre and the North Equatorial Current (NEC) (Figure 1). To the southeast, the North Brazil Current (NBC) runs northwestward along the coast of South America. Upon encountering the North Equatorial Countercurrent (NECC) from the opposite direction, the NBC bifurcates to the east forming a seasonal retroflection zone at about 6°-8°N, from late summer to

early winter. During intense retroflection events, antcyclonic eddies, known as NBC eddies, break away from the meander and evolve off the South American coast, along the 200 m isobath towards the Caribbean Archipelago, which acts as a natural dynamic "perforated" barrier. Following the NBC, the Guiana Current (GC) and then the South Caribbean Current (SCC) continue along the coast flowing north-westward. The Caribbean Counter Current (CCU) opposes the SCC and GC at depth. The instability of the SCC generates anticyclonic eddies in the southern Caribbean Sea, west of the Caribbean Archipelago,

which also evolve off the North American coast, northwestward to the Yucatan Strait.

Fratantoni et al. (2006) specifically study the characteristics of NBC eddies from the trajectory of floats and drifters deployed in the south-east of the Caribbean Archipelago, close to the NBC retroflection, at different depths. These eddies trap and advect warm, low-salinity, nutrient-rich water suitable for primary production from the mouth of the Amazon. They have a lifespan of about 2-3 months. For their surface component (0-200m), their diameter is of the order of 200 km for a rotation speed of 9

to 14 days. After they hit the island of Barbados at about 60°W and 13°N, several scenarios were observed: before dissipating, they either get stuck at the level of Barbados and remain stationary for a while, or move northwards along the Caribbean Archipelago, up to about 18°N. According to laboratory studies by Cenedese et al. (2005), a NBC eddy could also split into a





cyclone/anticyclone dipole of smaller diameter as it passes between the two islands of Saint Vincent and Barbados. Cenedese (2002) shows through his laboratory experiments that dissipation occurs when the obstacle/eddy diameter ratio is greater than

0.6. The different end-of-life scenarios for NBC eddies show that the Caribbean Archipelago clearly constitutes a dynamic barrier for these eddies which cannot cross it. However, Richardson (2005) found that the interaction between the NBC eddies and the Caribbean Archipelago could favor the generation of SCC eddies on the western side of the arc in the Caribbean Sea.

Here we show the benefit of the CAR36 high-resolution solution compared to that of GLO12 and focus particularly on the

evolution of a surface NBC eddy, its interaction with the islands, its potential effect on the instability of the SCC west of the Caribbean Archipelago and its influence on the Sargassum drift. Through the explicit resolution of such processes we expect CAR36 regional system to bring more accuracy compared to the lower resolution GLO12 global system and as such improve the forecast of Sargassum drift and beaching on the Caribbean Archipelago, by the Météo-France MOTHY model.

## 3.  Description of the GLO12 and CAR36 systems

### 3.1.  The global analysis and forecasting system GLO12

The global operational system GLO12 was developed at Mercator Ocean international for the Copernicus Marine Service (CMEMS, http://marine.copernicus.eu/) and simulates physical ocean variables (e.g., temperature, salinity, sea surface height and velocity) and sea ice variables from 2007 onwards. A full description of the system components is available in Lellouche et al. (2018). We just summarize here the main features listed in Table 1. The physical configuration is based on a 1/12° tripolar

grid (Madec and Imbard, 1996), with 50 vertical levels of decreasing resolution from 1 m at the surface to 450 m at the bottom, including 22 levels within the upper 100 m. The system GLO12 uses version 3.1 of the Nucleus for European Modelling of the Ocean model (NEMO; Madec et al., 2008). At the surface, the model is driven by atmospheric analyses and forecasts obtained from the European Centre for Medium-Range Weather Forecasts-Integrated Forecast System (ECMWF-IFS) at 3-hr resolution.

Altimetry data, in situ vertical temperature and salinity profiles and satellite sea surface temperature are assimilated using a reduced-order Kalman filter derived from a SEEK filter (Brasseur and Verron, 2006), with an assimilation cycle of 7 days (Lellouche et al., 2013). In addition, a 3D-Var scheme provides a correction for slowly evolving large-scale biases for temperature and salinity.

The GLO12 system outputs are available on a regular grid on the CMEMS portal in daily averages but also in hourly averages

for some two-dimensional fields, such as SST, SSH or surface current. These outputs are documented and validated. Currently, the daily two-dimensional currents under the Ekman layer, provided specifically to Météo-France by Mercator Ocean, are used to force the MOTHY drift model.



### 3.2.   The regional forecasting system CAR36

The CAR36 ocean model ingredients are based on those of the CMEMS regional IBI system at 1/36° (Iberia Biscay and
Ireland; Sotillo et al., 2015), transposed to the Caribbean Archipelago region. The main features of CAR36 system described
in detail below, are summarized in Table 1. CAR36 system uses version 3.6 of the NEMO ocean model (Madec et al., 2008).
The ORCA three-pole grid (Madec and Imbard, 1996) is considered with a horizontal resolution of 1/36°, i.e. approximately
3 km in the Caribbean Archipelago zone (64.25°W, 7.89°N / 54.17°W, 20.08°N). The vertical discretization of 50 levels is
identical to that of GLO12. A partial step parameterisation (Adcroft et al., 1997) was also chosen. Tracer advection is calculated
with the QUICKEST scheme developed by Leonard (1979) and limited by Zalesak (1979). This third-order scheme is well
suited to the high resolution used in CAR36 and to the modelling of well-marked fronts, typical of coastal environments. The
sub-mesh lateral mixing is parameterised according to horizontal biharmonic operators for momentum and tracers. A vector
invariant form of the momentum equations and the energy-enstrophy discretisation of the vorticity terms are used (Barnier et
al., 2006). The model uses a non-linear explicit free surface (Shchepetkin and McWilliams, 2004) to correctly simulate fast
external gravitational waves such as tidal waves. All vertical layers in the model are regridded vertically at each time step to
account for the varying height of the fluid. The vertical mixing is parameterised according to a k-ε model implemented in the
generic form proposed by Umlauf and Burchard (2003) and including the mixing induced by the breaking of surface waves
(Reffray et al., 2015). The bottom turbulent variables are fixed at their molecular values and are modified according to the
mixing induced by the internal tides not resolved by the model following the parameterisation of Lavergne et al. (2016). The
lateral friction condition is a partial slip condition. As for GLO12, the bathymetry used in CAR36 is a combination of ETOPO1
(Amante and Eakins, 2009) and GEBCO8 (Becker et al., 2009) databases. These very high-resolution data (respectively: 1 arc
minute and 30 arc seconds) are directly interpolated on the 1/36° ORCA grid. Ten grid points around the CAR36 domain
ensure the transition between the bathymetry of the regional model and that of the forcing global model. CAR36 is forced by
hourly atmospheric fields (10 m wind, surface pressure, 2 m temperature, relative humidity, precipitation, short- and long-
wave radiative fluxes) provided by ECMWF empirical IFS formulae (ECMWF, 2014; Brodeau et al., 2017) and used to
calculate sensible latent heat fluxes, evaporation and surface wind stress. Surface atmospheric pressure is considered here,
allowing for inverse barometer effects. The higher time resolution as well as the inverse barometer effects make these
atmospheric forcings more suitable to the extreme events on an area regularly hit by hurricanes. Solar penetration is
parameterised according to a five-band exponential scheme (considering UV radiation) as a function of surface chlorophyll
monthly concentrations. For the baroclinic mode, the open ocean boundaries are specified in temperature, salinity, zonal and
meridional velocities and sea surface height coming from daily GLO12 outputs. A relaxation band is used along the boundary.
For the barotropic mode, a Flather (1976) condition is applied to the boundaries.  Following the CMEMS IBI system
configuration, CAR36 takes into account tidal forcing also, even if the Caribbean area has a micro tidal range reaching between
10 and 20 cm (Kjerfve, 1986). The sea surface height and currents associated with the tide is prescribed to the FES 2014 global
ocean tide atlas data  (Lyard et al., 2020, Carrere et al. 2015) at the CAR36 boundaries.  FES2014 is the last version of the



FES (Finite Element Solution) tide model based on the tidal barotropic equation T-UGO model in a spectral configuration, and assimilated by long-term altimetry data (Topex/ Poseidon, Jason-1, Jason-2, TPN-J1N, and ERS-1, ERS-2, ENVISAT) and tidal gauges. Eleven tidal harmonics are considered (M2, S2, N2, K1, O1, Q1, M4, K2, P1, Mf, Mm): the four principal components M2, S2, N2 and M4 were recalculated on a one-year CAR36 simulation from a harmonic analysis of NEMO,

before being reapplied to the boundaries. This method allows the tide to fit the high-resolution bathymetry of the regional model, while the other seven harmonics are directly interpolated to the CAR36 grid.

CAR36 does not directly assimilate observations but includes a Spectral Nudging method applied to the prognostic variables: temperature T, salinity S and speed current components U and V. Following the principle explained by Von Storch et al. (2000) where a regional climate model was considered, in this present case the Spectral Nudging allows to constrain the large-scale

and low-frequency solution of CAR36 towards the solution of the global lower resolution GLO12 system which does assimilate directly observations. In particular, the idea is to reposition the structures of dimension greater than or equal to the mesoscale in accordance with the observations. The small-scale and high-frequency solution of CAR36 is left free. Unlike the method adopted by Von Storch et al. (2000) consisting in considering a large scale spectral domain and a small scale one defined by a spectral analysis,    the low and high resolution signals in space and time has been separated by applying a spatial filter with a

cutoff wave-length of 50 km as well as a weekly mean . Indeed, 50 km and one week correspond to the minimum characteristic length and period of the mesoscale at this latitude. In practice, the spectral nudging CAR36 is applied through an increment computed according to equation (1). It is defined as the difference between the analysed prognostic variable of GLO12 ($X_{GLO12}$) and the predicted one of CAR36 whose fine spatial scale has been filtered ($X_{CAR36\_filt}$). A weekly average ($<\ >$) is applied in order to keep only time scales greater than or equal to the mesoscale.

$$\delta = <X_{GLO12} - X_{CAR36\_filt}> \qquad\qquad (1)$$

As for GLO12, this increment is integrated weekly following the IAU principle (Blum et al, 1996; Benkiran and Greiner, 2008). It is applied spatially in the open ocean (where the bottom exceeds 200 m), 30 km from the coast and over layers of depth equivalent to half the water column. CAR36 therefore also provides forecasts and analyses from Spectral Nudging as it is shown in the **Figure 2**. In this paper we consider only analyses.

To better represent of fine-scale ocean dynamics, in addition to the resolution of the model mesh, bathymetry and coastline, some high frequency processes are also considered such as the forcing by hourly atmospheric data or the forcing by the tide. Particular attention has therefore been paid to the way in which the regional model must be driven by the global model via its initialisation and its open sea boundary conditions. In contrast to GLO12, the relatively low numerical cost of CAR36 has made it possible to perform several sensitivity experiments to the above-mentioned high-resolution and high-frequency

parameters on the one hand, and to the constraints of GLO12 on the other.



## 4    Validation Methodology CAR36

### 4.1  Metrics

The validation methodology applied to the GLO12 and CAR36 analyses (described in Lellouche et al, 2018) relies on applying a well-defined set of metrics (Oceanpredict metrics https://oceanpredict.org/) distributed in four main classes. These classes

include eyeball verification of oceanic features (as an example of class 1), comparison to high-frequency observational time series (as an example of class 2), comparison to published estimates of integrated quantities (for instance eddy kinetic energy or transports as an example of class 3) and accuracy and forecast skill estimates in the observations' space (class 4). Applying the full set of metrics ensures consistency, robustness, and accuracy of operational oceanography products from both the physical and statistical point of view.

Lagrangian metrics, and feature based (eddy detection) skill scores complement this set of metrics with the goal of providing fitness-for-purpose information. Highlights of the validation of CAR36 are provided in the following section, focusing on the main differences and potential added value relative to GLO12. Showing the added-value of a high-resolution regional model relative to a lower resolution global model remains a challenge. Namely, regional models better resolve the fine-scale structures but do not necessarily place them in the right place, especially since the small scales are not observed in the area of interest to

an extent which would allow constraining them with data assimilation. As a consequence, forecast scores of a high resolution regional forecasting system may not improve upon coarser forecasting system ones (or could even degrade them) depending on the scales resolved by the observations at hand.

In the following section large scale or so-called macroscopic validation scores over the year 2019 show the consistency between the forecast initial conditions produced by the two systems, while diagnostics dedicated to the validation of fine-scale

processes demonstrate the added value of CAR36. The fitness-for-purpose of CAR36 is illustrated with a case study of a surface NBC eddy, the different stages of its evolution and the potential impact on the instabilities of the South Caribbean Current (SCC) west of the Caribbean Archipelago and Sargassum aggregation zones as detected by satellites through the AFAI (Alternative Floating Algae Index) for instance. Given the limited amount of available observational data, the realism of the simulated processes is also based on literature descriptions.

### 4.2  Observations databases for validation


In order to evaluate the CAR36 and GLO12 simulations over the year 2019, five observations databases were used and extracted over the region and period of interest.

The daily mean absolute dynamic topography (MADT) from AVISO altimetry products – hereafter called AVISO SSH L4– available on the Copernicus Marine Service (CMEMS) portal (https://doi.org/10.48670/moi-00149) is used for statistical

diagnostics such as maps of bias, correlation and RMS deviation of the models from the observations. This type of diagnosis requires gridded data with no missing values outside the continents. MADT data are of the L4 type based on data measured along the tracks of altimetry satellites interpolated on a regular 1/4° grid (~ 25 km) following an optimal interpolation filling





in the missing values between the tracks, thus obtaining a homogeneous field. This relatively low-resolution product resolves structures down to the mesoscale.

As a preamble to their intercomparison, the effective resolution of AVISO SSH L4 maps, CAR36 and GLO12, which defines the scales that the analyses actually resolve, is illustrated in **Figure 3** with a spectral analysis performed over the year 2019 and within the CAR36 domain. As expected, CAR36 is more energetic than the 2 other products. The tail of the spectrum expands with the resolution as the dissipation zone is pushed further towards the fine scales. The dissipation zone is determined at the point where the spectrum starts to reach a threshold towards the small scales. The limit at which the spectrum becomes

dissipative is used to define the effective data resolution. This limit is reached for a wavenumber of about $10^{-1}$ km$^{-1}$, i.e. an effective resolution of about 10 km for CAR36, compared to a wavenumber of about $4.10^{-2}$ km$^{-1}$, i.e. an effective resolution of 25 km for GLO12. As an indication, the effective resolution of the AVISO L4 SSH reaches 50 km, thus allowing a good part of the mesoscale signal to be resolved. It should be noted that the spectral analysis is often carried out by latitude band since the effective resolution depends on the latitude and more precisely on the Rossby deformation radius as a function of the

latitude and depth. This radius decreases especially towards high latitudes and in shallower regions such as the Mediterranean. Here the CAR36 domain is located in the low latitudes and its extent is quite small. The analysis over the whole domain therefore seems reliable.

The AVISO SSH L4 product is also used to detect and to track eddies. The use of AVISO SSH L4 at ¼° can be a real limitation in terms of comparing simulated and observed eddy detection. Stegner et al. (2021) show via an Observing System Simulation

Experiment that artefacts on eddy detection can be due to the optimal interpolation applied between altimeter tracks to obtain the AVISO L4 SSH. Two cases of error can occur: either an unrealistic eddy is detected, or a single eddy is detected when several actually exist. It seems important to evaluate the limitations of these data, however in our study we focus on one particular NBC eddy whose diameter is large enough and resolved by the AVISO SSH L4. The detection and tracking tool is then applied to the evolution of a fully developed NBC eddy before it dissipates on arrival at the Caribbean Archipelago.

Daily sea surface temperature (SST) data from the UK Metoffice OSTIA products available on the CMEMS portal (https://doi.org/10.48670/moi-00165) are used as the AVISO L4 SSH for statistical diagnostics only. Like the previous data, this L4 SST is homogeneously interpolated on a regular 1/20° grid with an effective resolution of 10 km (Donlon et al, 2012). Missing values due to the presence of clouds are filled in.

Daily L3S (Super-collated) SST data from IFREMER (Institut Français pour la Recherche et l'Exploitation de la MER)

products available on the CMEMS portal (https://doi.org/10.48670/moi-00164) are used for both qualitative validation diagnostics – such as visual comparison of SST maps at given dates for monitoring NBC eddies trapping warm waters – and for more quantitative assessment – for instance mean differences over the area. This SST is interpolated on a regular 1/20° grid (~ 5 km) leaving missing values at cloud locations and as such avoids artefacts related to the optimal interpolation.

Lagrangian trajectory data from surface drifters produced by IFREMER and CLS (Collecte Localisation Satellites) and

available on the CMEMS portal (https://doi.org/10.17882/86236) allow us to compare the distribution of the trajectories





deduced from the Eulerian velocity of CAR36 and GLO12 colocated at real drifters over the whole 2019 period and the whole CAR36 domain.

Daily sargassum detection data (Berline et Descloitres, 2021) from Alternative Floating Algae Index method (Wang and Hu, 2016) and averaged by 5x5 km pixel are used to determine the areas of Sargassum aggregation over the CAR36 domain. These

allow qualitative validation of the CAR36 and GLO12 model dynamics.

## 5  Results

### 5.1  Macroscopic validation

The consistency of the CAR36 and GLO12 sea level is evaluated by intercomparing with L4 AVISO maps as reference observations. Bias, correlation and RMS deviation maps are computed on the observations grid (**Figure 4**). A Shapiro spatial

filter has been previously applied to the simulated data before being interpolated on the coarser observed data grid, in order to avoid sampling errors and compare the scales resolved by the observation data (> 50 km). As expected, the improvement of CAR36 is not obvious. However, the bias of CAR36 seems to be slightly reduced along the Caribbean Archipelago, an area dominated by sub-mesoscale structures and corresponding to the dissipation region of NBC eddies. First, the observational data used have low resolution (1/4°) and are therefore more suitable for large-scale model validation. Consequently, the local

statistical deviations of the regional model from the observations can be equivalent or even degraded compared to those of the global model. Then, this type of analysis does not necessarily show the improvements of the regional model compared to the global one: the finer structures are not constrained by observations and if they are better resolved by the regional model, they are not necessarily well positioned despite the spectral nudging applied on the large scale. However, the good consistency between CAR36 and GLO12 validates the technical choice of a spectral nudging applied to CAR36 with respect to a more

complex data assimilation scheme.

The same type of analysis and intercomparison is performed for SST using the OSTIA SST analysis (**Figure 5**). Although minimal, the differences in results between CAR36 and GLO12 are more noticeable compared to the previous diagnostics on SSH. The biases and RMS deviations are generally lower for CAR36.

These analysis have the drawback of evaluating the models in relation to the observations in a local manner, i.e. by applying

statistics point by point on the grid of observation data over the entire period under consideration.

A non-local approach can complete these diagnostics.

The velocity field of the CAR36 is assessed using the surface drifter data from both a Lagrangian and an Eulerian perspective. The gridded Eulerian velocity of the systems CAR36 and GLO12 has been colocated at real surface drifters' position over the CAR36 domain and the 2019 period.  Surface current speed (U,V) have been interpolated from gridded model data on each

position and time of the drifters with a time and space interpolation. The space interpolation on position is performed with inverse squared distance weighting interpolation over closer 4 adjacent grid cells of the gridded model. The time interpolation is linear. A representation in the form of a current rose, as shown in **Figure 6**, makes it possible to compare the distribution of





the trajectories deduced from the colocalization previously describe with those of the real drifters, according to the direction and speed of movement of the drifters. For the 39 drifters available over the region and the period, the distribution of the CAR36 deduced trajectories is close to that of the real drifters. Although the average direction of the GLO12 deduced trajectories  indicated by the mixed line on the current rose, corresponds better to that of the real drifters, the percentage of west-north-west trajectories is largely overestimated. In this type of analysis, the contribution of the regional model seems clearer. This diagnosis seems to be better suited to the validation of high-resolution regional models.

**5.2  Focus on a particular NBC eddy and the associated processes**

This macroscopic assessment is here complemented by more qualitative diagnostics that focus on the contribution of high resolution to the representation of fine-scale processes such as the evolution of NBC eddies and their potential impact on the ocean dynamics of the Caribbean sea, west of the Caribbean Archipelago. We consider the evolution of one NBC eddy, generated in January 2019.

In order to distinguish the different evolution scenarios of NBC eddies depending on whether it was simulated by CAR36 or GLO12, a Lagrangian particle seeding experiment was performed within such an eddy that just detached from the NBC retroflection in January 2019.    Virtual particles were released and tracked using the OceanParcels tool (https://oceanparcels.org/) together with the Eulerian surface currents simulated by the two models. A large domain including the CAR36 domain and the retroflection zone is considered. Initially, in the core of a newly generated NBC eddy (5°N-8°N/49°W-52°W) located in the south-east corner of the wide domain and outside the CAR36 domain, 4x4 virtual floats were seeded. The trajectories of the virtual Lagrangian floats are plotted in **Figure 7** for GLO12 and CAR36 over the chosen extended region. We note that the trajectories differ especially when the eddy arrives on the Caribbean Archipelago: in the case of GLO12, the eddy seems to dissipate as it passes the island of Barbados (located at about 59.5°W and 13.2°N) whereas the eddy simulated by CAR36 seems to want to pass the island of Barbados to progress northwards along the Caribbean Archipelago, its diameter having been reduced. As described above, these two evolutionary scenarios are possible and have been observed by Fratantoni et al. (2006). It remains to be determined which of the two scenarios is the most realistic for the period studied.

Since the NBC eddy has a SST signature, these evolution scenarios are then compared with satellite SST maps (SST L3S described above). In **Figure 8**, where again the CAR36 domain is considered, GLO12 and CAR36 simulate the entry of the NBC eddy in the south-east corner in an equivalent way. On 26 February 2019, the warm water wrap around the simulated eddy is found in the observed L3S. At this stage of evolution where the NBC eddy is fully developed with a radius of the order of 200 km, the 1/12° resolution may be sufficient and the use of a higher resolution model is not necessarily justified, at least to resolve this type of mesoscale structure. On 17 March 2019, the eddy, having trapped the warm waters, arrives over the Caribbean Archipelago south of the island of Barbados. The diameter of the eddy obtained by GLO12 is then much larger than the observed one, while that of CAR36 is closer to the observed one, but with a colder SST signature. The observed eddy is also smaller and lies further west along the arc. On 25 March 2019, the eddy simulated by GLO12 dissipates and gradually





loses its SST signature, while the eddy from CAR36 still retains its SST signature and is slightly distorted. As for the observations, warm waters spread out along the Caribbean Archipelago up to 16°N. This analysis still does not allow to discriminate which system gives the most realistic results, especially when the eddy hits the Caribbean Archipelago just before it dissipates.

The eddy detection and tracking tool (https://py-eddy-tracker.readthedocs.io/en/stable; Mason et al., 2014), applied to the SSH field, allows to refine our analysis regarding the representation of the evolution of the eddy studied until its arrival on the Caribbean Archipelago just before its dissipation, according to the 3 types of data considered from GLO12, CAR36 and AVISO (SSH L4).

The eddy detection method is based on an SSH criterion: it first detects closed contours of the same SSH every 2 cm of SSH

interval. These iso-contours are retained by the method if the error on the shape does not exceed 70%. This error is characterized as the ratio of the difference between the area of the detected contour and the contour of the nearest optimal circle (greyed area on the left panel of **Figure 9**), to the area of this optimal circle (greyed area on the right panel of Figure 10). The minimum area of the discs is set to 5 pixels (of the considered data grid) and the maximum area to 10,000 pixels. For this, the SSH fields are first filtered spatially for all 3 products (GLO12, CAR36 and AVISO)  so to target the mesoscale range:

spatial structures smaller than 50 km – i.e. the effective resolution of AVISO, **Figure 3** – and larger than 500 km are removed. The tracking of eddies over time is based on the area overlap between detected eddies of two consecutive days. For an eddy at time t, we look for the next day (t+δt) for eddies that have an overlap rate greater than 2%. The overlap rate is defined as the ratio of the intersection of the areas of the eddies to the union of the two areas. If several eddies share an area at time t+δt, the one with the highest overlap rate is considered. Such method does not keep account of eddy splitting and merging, which may

lead to errors in estimating eddy lifetime and trajectory of the eddies. The tracking method is subject to two parameters: the minimum eddy-lifetime, here set at 10 days, and the fictitious detection time corresponding to the lapse of time during which the same eddy would not have been detected, fixed here at 3 days. The choice of eddy detection and tracking parameters applied here from the eddy tracker tool is based on the work of Pegliasco et al. (2020).

From the eddies detected and tracked over the year 2019, the eddy tracker tool makes it possible to identify a particular eddy.

The NBC eddy studied so far, its trajectory, its characteristics could be extracted.

Particular focus was set on investigating the dynamics of one NBC eddy identified in the January 2019 entering the CAR36 study area in all 3 products (GLO12, CAR36 and AVISO). Figure 10 compares the trajectory of this eddy in all 3 products, from its entry into the domain to the east, its impact on the Caribbean Archipelago and finally its dissipation. The CAR36 trajectory (green) is closer to AVISO (red) than the GLO12 trajectory (blue), especially near the islands, suggesting that the

NBC eddy evolution scenario obtained by CAR36 (as seen earlier in the Lagrangian particle seeding experiment applied to the same eddy (see **Figure 7**) and consisting of a stationary eddy over Barbados) seem more realistic. The shape of the end-of-life eddy from CAR36 also seems to be more consistent with that obtained from observations. The lifetime of the eddy simulated by GLO12 and CAR36 is however overestimated by 13 and 12 days respectively. It should be noted that the limitation of the eddy tracking method with respect to splitting and merging phenomena does not really concern the NBC eddies before they



pass the island of Barbados and dissipate. Regarding splitting, laboratory experiments by Cenedese et al. (2005) show that a NBC eddy could split into a smaller diameter cyclone/anticyclone dipole after passing between the two islands of St Vincent (located at about 61.2°W and 13.2°N) and Barbados, but not before. Regarding merging, fully developed NBC eddies are the most energetic eddies in the region and could potentially absorb other nearby smaller eddies without affecting its detection and tracking. Furthermore, the effective resolution of the AVISO L4 SSH reaching 50 km in the region is not limiting either.

The type of diagnostic in Figure 10 is therefore relatively reliable for this case study and shows the contribution of the regional model to better representing the evolution of a NBC eddy, in particular when approaching the Antillean islands.

We are now interested in the dissipation phase of the NBC eddy, the associated energy transfer across the Caribbean Archipelago and the possible impact on the instability of the SCC current and the formation of the SCC eddy. From climatologies computed with AVISO data and a 1/12° resolution model over a long period from 1993 to 2009, Jouanno et al.

(2012) had already noticed an east-west eddy kinetic energy (EKE) transfer in a wider region encompassing the Caribbean Archipelago and the entire Caribbean Sea. **Figure 11** presents Hovmöller diagrams (longitude/time, for a latitude range from 7.89°N to 20.08°N) of EKE calculated from the geostrophic velocities deduced from AVISO's L4 SSH and the total velocities of GLO12 and CAR36. For the three types of data, oblique lines of higher energy reflecting the transfer of energy from east to west over the region under consideration, especially in the first part of 2019, with a maximum of energy as it passes the

islands of the Caribbean Archipelago. Note that the vertical lines correspond to the concentration of EKE, probably due to the shear stress between the islands and the resulting stronger current speed perturbations. The westward propagation of EKE is more marked in the case of CAR36. For a better analysis of this energy transfer, the relative vorticity field is diagnosed over the period of evolution of the studied NBC eddy from its entry into the domain until its dissipation, which allows distinguishing the fine-scale structures emerging on the Caribbean Sea side and the way they are resolved by the GLO12 and CAR36 models.

Conventional observational data used so far do not cover the sub-mesoscale range, thus other datasets are needed to validate the small scales reproduced by CAR36. For this purpose, the 5 km resolution Sargassum detection data is used with Sargassum aggrgations superimposed to CAR36 simulated vorticity field to verify the correct positioning of fine structures (**Figure 12**). Assuming that in this region of high mesoscale and sub-mesoscale activity, the Sargassum aggregation process is primarily related to fine-scale ocean dynamics rather than to wind , this comparison to Sargassum aggregates detections illustrates the

fitness-for-purpose of CAR36 and GLO12 for this specific case study. **Figure 12** shows the different stages of evolution of the NBC eddy in terms of relative vorticity. The green pixels on the maps represent the detected Sargassum. Given the significant cloud cover masking the detection, a cumulative sum over 3 days centered on the day was carried out. On february 26th, in agreement with the previous results, the fully developed eddy entering the domain is simulated in the same way for the 2 models. Some of the detected Sargassum appears to be trapped in the periphery, in the eddy winding. On March 17th, the eddy

approaches the Caribbean Archipelago south of the island of Barbados. The smaller diameter of the eddy simulated by CAR36 seems closer to reality: the coincidence between the alignment of the sargassum and the outer contours of the eddy simulated by CAR36 is notable, suggesting that the diameter of the eddy obtained by GLO12 is overestimated at this stage. On March 25th the eddy starts to dissipate over Barbados and filamentary structures intensify west of the Caribbean Archipelago. These





structures, such as the filament centered at about 14°N, appear to be better resolved by CAR36. The detection of sargassum for this date is not clear and does not allow the simulations to be validated. On April 13$^{rd}$ , the eddy simulated by GLO12 is almost completely dissipated, while the eddy simulated by CAR36 still has a relatively circular shape at 16°N along the Caribbean Archipelago. Sargassum still seems to be trapped by this eddy. And to the west, in the Caribbean Sea, an SCC eddy is developing, centered at about 62°W and 13°N, which is clearly visible in the CAR36 simulation, unlike GLO12. Sargassum also appears to be trapped in this eddy. The formation of the SCC eddy potentially favored by filaments generated or intensified during the dissipation of the NBC eddy would be consistent with the observations of Richardson (2005).

## 6   Conclusion

A regional high-resolution ocean forecasting system at 1/36° (CAR36) over the Caribbean Archipelago was developed and put into operation for a concrete application of short-term Sargassum drift forecasting. The ocean currents from CAR36 are provided to Météo-France to feed the MOTHY drift forecasting model. These forecasts are invaluable for alerting the local authorities in Guadeloupe and Martinique of a massive Sargassum stranding on the beaches.

The objective of the work presented in this paper was to show the contribution of the CAR36 forecasting system compared to the coarser global system GLO12 at 1/12° with respect to the fine-scale ocean dynamics of the Caribbean Archipelago region. The year 2019 was considered and all results concerned this period. In addition to the high resolution, CAR36 also takes into account high frequency forcing such as the inverse barometer effect, hourly 1/10° IFS atmospheric data from the ECMWF and 11 tidal harmonics from the 2014 FES atlas data. CAR36 is not constrained directly by the data available in the region but by the large-scale solution of the GLO12 system which assimilates satellite and in situ observations. This constraint follows the "spectral nudging" method and particularly allows to make corrections to structures of scale greater than or equal to the mesoscale. The boundary conditions consistently come from GLO12, CAR36 thus can be seen as a local refinement of the GLO12 forecasting system with a so-called one-way nesting technique. The validation methodology carried out here to highlight the benefits of the regional model on local dynamics consists of first establishing quantitative statistical diagnoses with respect to the observations over the entire region and period considered for each of the two models GLO12 and CAR36, and then completing these diagnostics with a more qualitative study of a local fine-scale physical process such as the evolution of an NBC eddy. The power spectral density diagnostic over the region and period gives an estimate of the effective resolutions of the two models, i.e. the scales potentially resolved by GLO12 and CAR36, namely respectively 25 and 10 km. The bias, correlation and RMS deviation maps of SSH versus AVISO altimeter data did not show significant differences between GLO12 and CAR36. The low resolution of AVISO L4 SSH data re-gridded by optimal interpolation on a 1/4° regular grid does not seem suitable for validating a regional model through this type of diagnostic. The same diagnostic on the SST using the OSTIA L4 SST shows a slight decrease in bias and RMS deviations for CAR36. This diagnostic based on local statistics comparing point by point the simulated data to the observed data in the reference frame of the observations brings about the "double penalty" of a regional high resolution model:  the fine scale structures are better resolved but not well positioned.  A statistical





non-local analysis completes these results. The Eulerian velocity from CAR36 and GLO12 co-located at drifters show that the distribution of the directions of the trajectories simulated by CAR36 is clearly more in agreement with that of the observed trajectories reducing the overestimation of the west-northwest direction for GLO12. A focus is then made on the evolution of an NBC eddy from its entry into the studied region following its detachment from the NBC retroflection, until its dissipation

over the Caribbean Archipelago. Over a period of 3 months (from January to April) corresponding to the lifetime of the NBC eddy, a seeding experiment of Lagrangian particles within the eddy studied simulated by GLO12 and CAR36 makes it possible to distinguish two evolutionary scenarios. In the case of GLO12, the eddy does not go beyond Barbados and dissipates. In the case of CAR36, the eddy slows down, reduces, but goes beyond Barbados. It remains to be seen where the reality lies, given that each of these scenarios has already been observed by Fratantoni et al. (2006).  Qualitative comparison of the GLO12,

CAR36 and IFREMER L3S SST maps for fixed dates show very similar results when the fully developed NBC eddy moves westward. On the other hand, when it arrives south of Barbados, the characteristics of the eddy simulated by CAR36 seem to be closer to those of the observed eddy: the smaller diameter disc of trapped warm water is more in line with the observed disc. At this point the diameter of the GLO12 simulated eddy is largely overestimated. At the next evolutionary period, the CAR36-simulated eddy overtakes Barbados Island, deforming slightly and still retaining its SST signature, while the GLO12-

simulated eddy dissipates and gradually loses its SST signature. Concerning the observed SST, warm waters are spreading along the Caribbean Archipelago, further north. At this stage, the diagnostics presented do not yet allow us to discern, between CAR36 and GLO12, the most realistic evolution of the simulated eddy. To complement this analysis, a new eddy detection and tracking method based on the "py_eddy_tracker" tool (Mason et al, 2014) allowed, from the SSH field, to determine the trajectory and shape of the simulated and observed eddy during its evolution until its arrival on the island of Barbados, as long

as it could be detected before the dissipation phase. The trajectory and shape of the eddy simulated by CAR36 are better correlated with those obtained with the L4 AVISO SSH, especially towards the end of the evolution period in the vicinity of Barbados. Note that despite the smoothness of the L4 AVISO SSH, these data, whose effective resolution has been estimated at 50 km in the region, are still usable for this type of non-local diagnosis, considering eddies of the order of 200 km in diameter. The next phase of the evolution of the studied NBC eddy concerns its dissipation, the associated energy transfer

across the Caribbean Archipelago into the Caribbean Sea, the possible impact on the instability of the SCC current and the formation of SCC eddies. During this phase the development, filaments and interaction of sub-mesoscale structures can only be resolved by the regional model. During the dissipation of the NBC eddy, the westward transfer of kinetic energy across the Caribbean Archipelago is reflected in the Hovmöller energy diagram (longitude/time) by periodic oblique lines of maximum energy over the year 2019, and in all 3 data types: CAR36, GLO12 and AVISO. This transfer, which is much more pronounced

in the case of CAR36, is reflected in the relative vorticity maps by the development of filaments created between the islands of the Caribbean Archipelago. These filaments are finely resolved by CAR36 unlike GLO12. Consistent with the observations of Richardson (2005), their interaction with the SCC current appears to contribute to the creation of an SCC eddy, again for the CAR36 simulation. Recent Sargassum detection data are overlaid on the vorticity maps. In the region of high turbulent activity, the Sargassum aggregation zones appear to align with the contours of the meso- and sub-mesoscale structures (NBC





eddy, filaments and SCC eddy). Sargassum would act as a tracer in these fine-scale dynamics. Thus the detection of these algae would support the validation of the sub-mesoscale processes simulated by CAR36.

## 7    Prospects

On the application side, Météo-France performed sensitivity tests to CAR36 currents versus GLO12 on the Sargassum drift forecasting model MOTHY, and concluded, after comparison to drifting buoys, that the best short-term forecast scores

(performed in the past in 2019) were achieved when MOTHY, configured in full immersion "container" mode, was forced by the daily CAR36 currents averaged over the first 100 meters (not shown). These preliminary results must be completed by more significant statistical diagnostics and by using more observations in the region which will be available following a 500 planned Sargassum drift observation campaign next year.

We considered a reference simulation period in the past, the year 2019, and validated the analysis of this simulation. Such

work is preliminary and should be completed by forecasting' scores. These however can only be carried out when a sufficiently large number of forecast will be available from the early operational production of the CAR36 system. We focused on a situation of an evolving NBC eddy and how it was represented by the CAR36 and GLO12 models. The study of a larger sample of eddies, over a longer period of time, would make the results presented here more meaningful, by taking into account several evolutionary scenarios observed during the incidence of NBC eddies over the Caribbean Archipelago, such as: the dissipation

of the eddies or their progression northwards along the arc (Fratantoni et al, 2006) or their dissociation into a dipole during their passage between the islands of Saint Vincent and Barbados (Cenedese, 2005). Furthermore, only the surface eddy dynamics were considered, and monitoring the evolution of several NBC eddies vertically would allow us to verify the vertical dissociation phenomena of certain eddies ("cleavage") as observed by Fratantoni et al, (2006). The validation of forecasts, i.e. data from the CAR36 model not constrained by spectral nudging, could complete this work. The forecasts in the past, called

reforecasts, would be confronted with observations and analyses, in order to establish forecast scores. We have seen that the eddy detection and tracking tool "py_eddy_tracker" could be limited by the resolution of the SSH data used, such as the AVISO data. An evolution of the tool towards other detection criteria using higher resolution data such as the ocean color data or the future SWOT altimeter data would allow to obtain diagnostics more adapted to the validation of eddy structures simulated by a regional model.

It would been interesting to implement a CAR36 two-way AGRIF nesting zoom around Guadeloupe and Martinique islands in order to improve local sub-mesoscale processes and thus the MOTHY Sargassum drift forecasts as well as the beaching monitoring along the islands. A full data assimilation method by using regional data would have been able to replace the present spectral nudging method in order to constrain sub-mesoscale structures also. A next study is planned to test the sensitivity of CAR36 to the higher resolution atmospheric forcings' data from Météo-France AROME-ANTILLES forecasting

system. This regional system includes the Caribbean Archipelago in its domain and reach a 2.5 km horizontal resolution. Besides, AROME-ANTILLES is based on a non-hydrostatic model which resolves vertical acceleration and consequently



better represents hurricanes in this area. The hindcasts of CAR36 forced by ECMWF data described in this paper will be compare by the ones of CAR36 forced by AROME-ANTILLES for the same period 2019.

## 8 Data availability

All the native CAR36 simulation data used in this paper are available following this link:
https://doi.org/10.5281/zenodo.8386855

## 9 Author contribution

Sylvain Cailleau led the study presented in this paper, he processed the different observation and model data, developed and applied validation diagnostics. He wrote the manuscript with contributions from all co-authors. Laurent Bessières set up the
CAR36 regional forecasting system with the help of Guillaume Reffray and Jean-Michel Lellouche and he carried the CAR36 simulations out. Simon van Gennip, Charly Régnier, Marie Drevillon and Sylvain Cailleau supervised the study works of Léonel Chiendje and Flavie Dubost in the framework of their internship in Mercator Ocean international. The developments and outcomes of Flavie and Léonel about eddies' detection and tracking have been useful for the study presented here. Julien Temple-Boyer has developed, updated and maintained validation score scripts allowing the comparison of trajectories deduced
from simulated eulerian speed currents and drifters' ones, among other and which permit to perform the new non-local score of the figure 6.

Marc Tressol, Matthieu Clavier have implemented the operational production of CAR36. Léo Berline shared its knowledge about Sargassum drift and aggregation topics. He provided access to the satellite sargassum detection used in the figure 12.

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

| | **GLO12** | **CAR36** |
|---|---|---|
| **Ocean model** | NEMO 3.1 (Madec et al., 2008) | NEMO 3.6 (Madec et al., 2008) |
| **Domain** | Global | Antillean Arcs: 64.25°O,7.89°N / 54.17°O,20.08°N |
| **Horizontal resolution** | 1/12° (~ 9 km) | 1/36° (~ 3 km) |
| **Vertical resolution** | 50 z-levels: 1m at the surface, 450 m on the bottom | 50 z-levels: 1 m at the surface, 450 m on the bottom |
| **Advection Scheme** | For tracers: total decreasing variance (TDV) scheme (Lévy et al., 2001; Cravatte et al., 2007) + isopycnal lateral diffusion<br>For momentum: a conservation of energy and enstrophy scheme (Arakawa and Lamb, 1981) + biharmonic viscosity | For tracers: QUIKEST (Leonard, 1979 ; Zalesak, 1979) + biharmonic lateral diffusion<br>For momentum: a vector invariant form of the momentum equations and the energy-strophy discretization of the vorticity terms (Barnier et al., 2006) + biharmonic viscosity |
| **Free surface** | Free surface filtered from external gravity waves (Roullet and Madec, 2000) | Nonlinear explicit free surface (Shchepetkin and McWilliams, 2004) |
| **Convection scheme** | 1.5 order turbulent closure model (Blanke and Delecluse, 1993) | k-ε (Umlauf and Burchard, 2003) + mixing due to the wave breaking |
| **Bathymetry** | ETOPO1 (Amante and Eakins, 2009) and GEBCO8 (Becker et al., 2009) | ETOPO1 (Amante and Eakins, 2009) and GEBCO8 (Becker et al., 2009) |
| **Atmospheric forcing** | 3-hourly atmospheric forcing from the « bulk formulae » Large and Yeager (2009) | ECMWF hourly atmospheric forcing from the « bulk formulae »: ECMWF (2014); Brodeau et al. (2017) |
| **Solar penetration** | 3-band exponential pattern of surface chlorophyll concentrations (SeaWIFS monthly climatology) | 5-band exponential pattern as a function of surface chlorophyll concentrations (Copernicus ESA-CCI monthly product): OCEANCOLOUR_ATL_CHL_L4_NRT_OBSERVATIONS_009_090) |
| **Open boundary conditions** | No open conditions | For baroclinic mode: clamed boundaries clamed in T, S, U,V and SSH by the daily GLO12 data<br>For barotropic: Flather (1976) conditions |
| **Tidal forcing** | No tide | Tide prescribed to data (sea surface level and tidal current) FES 2014 (Carrere et al 2015), at CAR36 boundaries. 11 harmonics considered: M2, S2, N2, K1, O1, Q1, M4, K2, P1, Mf, Mm |
| **Data Assimilation** | Reduced order Kalman filter derived from SEEK (Brasseur and Verron, 2006) with a 7-day assimilation cycle (Lellouche et al., 2013) | Spectral Nudging |
| **Assimilated data** | Along-tracks altimeter SLA, in situ T and S vertical profiles, satellite OSTIA SST | Indirectly T, S, U, V from GLO12 |

**Table 1: Main features of the GLO12 and CAR36 systems.**






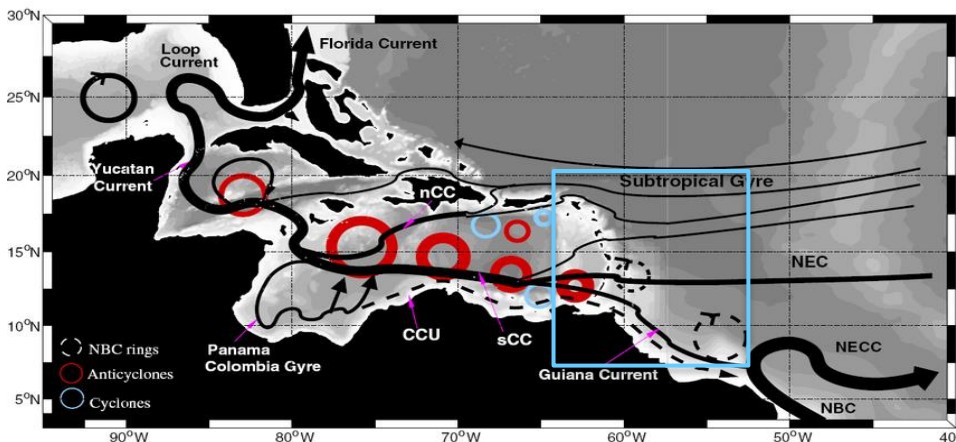

**Figure 1: Ocean circulation and local processes in the Caribbean region (from Jouanno et al., 2008). NBC: North Brazilian Current, NECC: North Equatorial Counter Current, NEC: North Equatorial Current, SCC: South Caribbean Current, CCU: Caribbean Counter Undercurrent, NCC: North Caribbean Current. The blue square delimits the domain of the CAR36 regional system.**

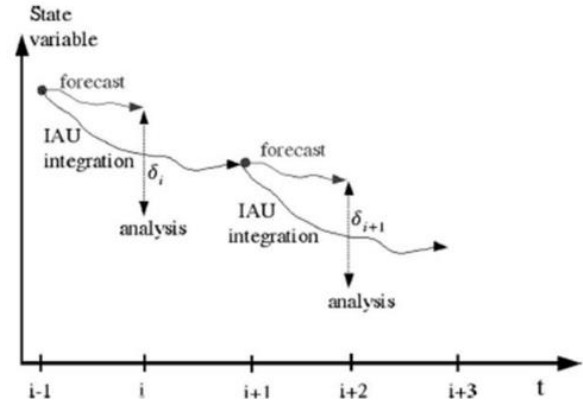


**Figure 2: IAU method from Blum et al., 1996. $\delta_i$ is the increment of the $i$ analysis cycle (here $i$ is weekly).**






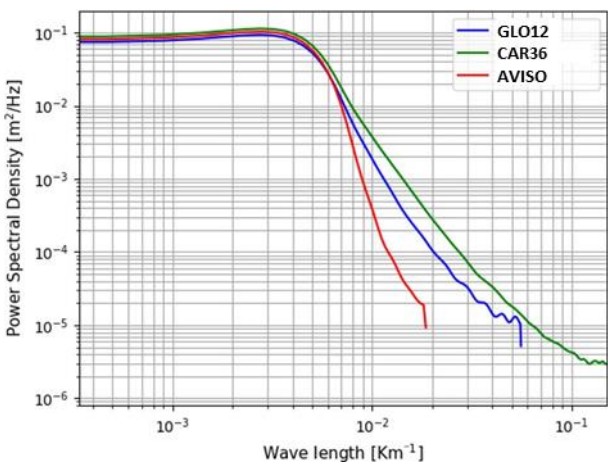

**Figure 3: SSH spatial spectra for CAR36 (green), GLO12 (blue) and L4 AVISO SSH observations (red) over the year 2019 and the CAR36 domain.**






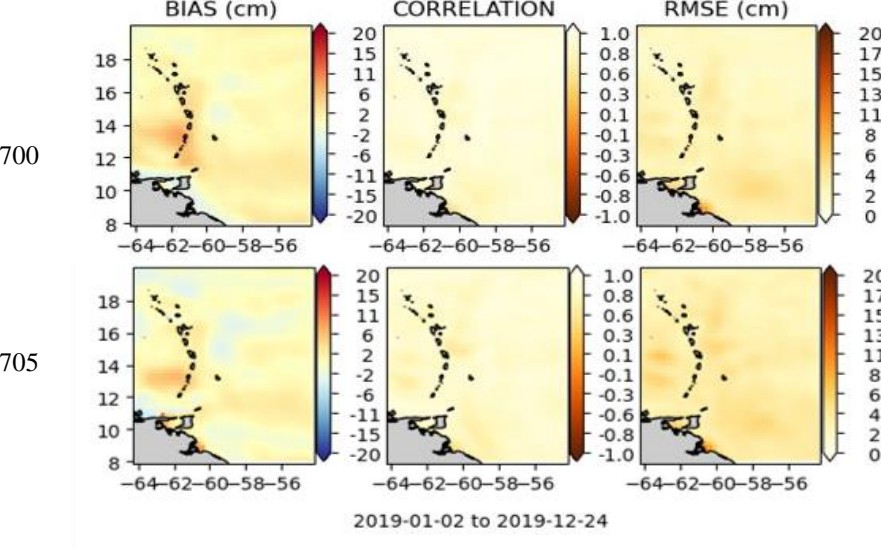

**Figure 4 : Maps of bias , correlation and RMSE of SSH between models and AVISO L4 SSH observations over the Caribbean arc region and over the period of 2019, for the GLO12 system on top and the CAR36 system on bottom.**








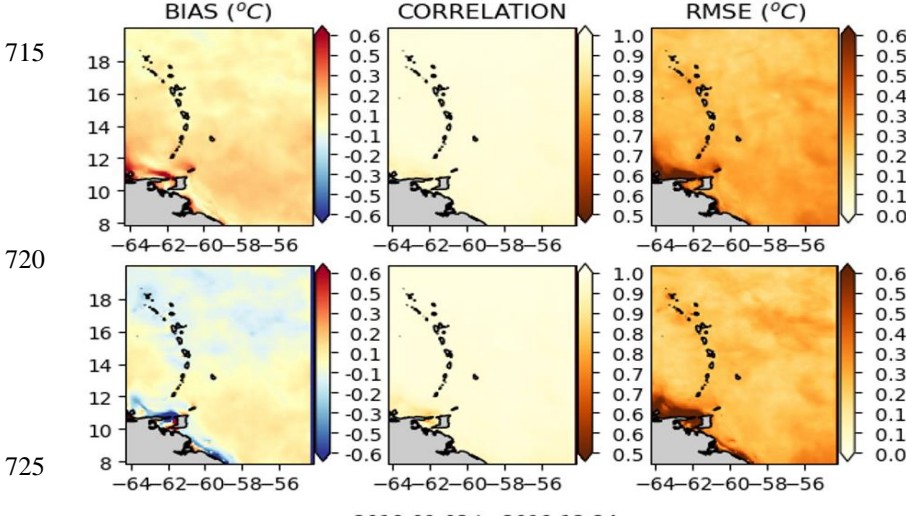

**Figure 5 : Maps of bias, correlation and RMSE between models and OSTIA L4 SST observations over the Caribbean arc region and over the 2019 period for the GLO12 system on top and the CAR36 system on bottom.**








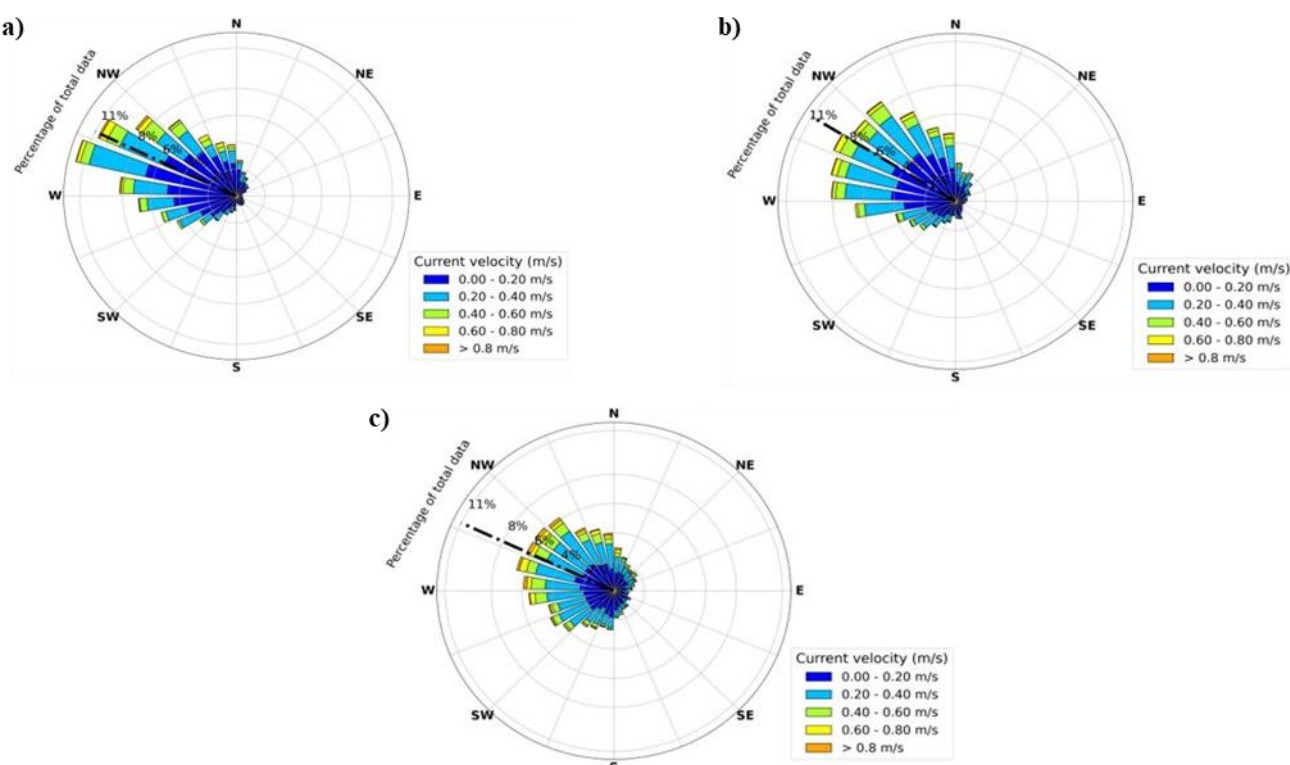

**Figure 6 : Surface current roses from the trajectories of real drifters (panel c) and along the same trajectories from the Eulerian velocity fields from GLO12 (panel a) and CAR36 (panel b) systems.**

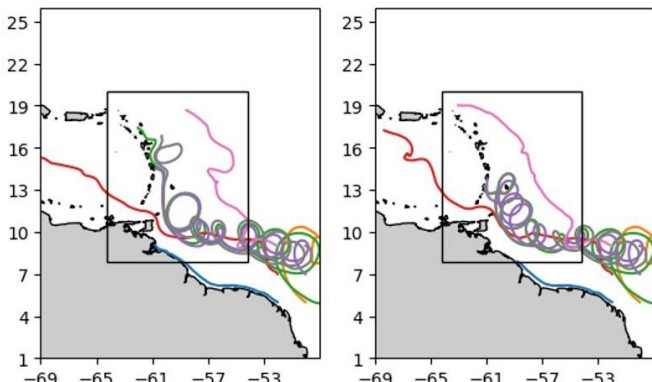

**Figure 7 : Virtual float trajectories over the extended area encompassing the Caribbean arc, for the GLO12 global model (left panel),**
**the CAR36 regional model (right panel). The black rectangle delimits the regional model domain. For both models, the initial**
**positions of the 16 floats were seeded within an NBC eddy, in the southeast corner (5°N-8°N/49°W-52°W block of 4x4 floats) outside**
**the regional model domain. The period chosen (01/01/2019 to 01/04/2019) corresponds to the evolution of the NBC eddy from its**
**birth to its end of life.**




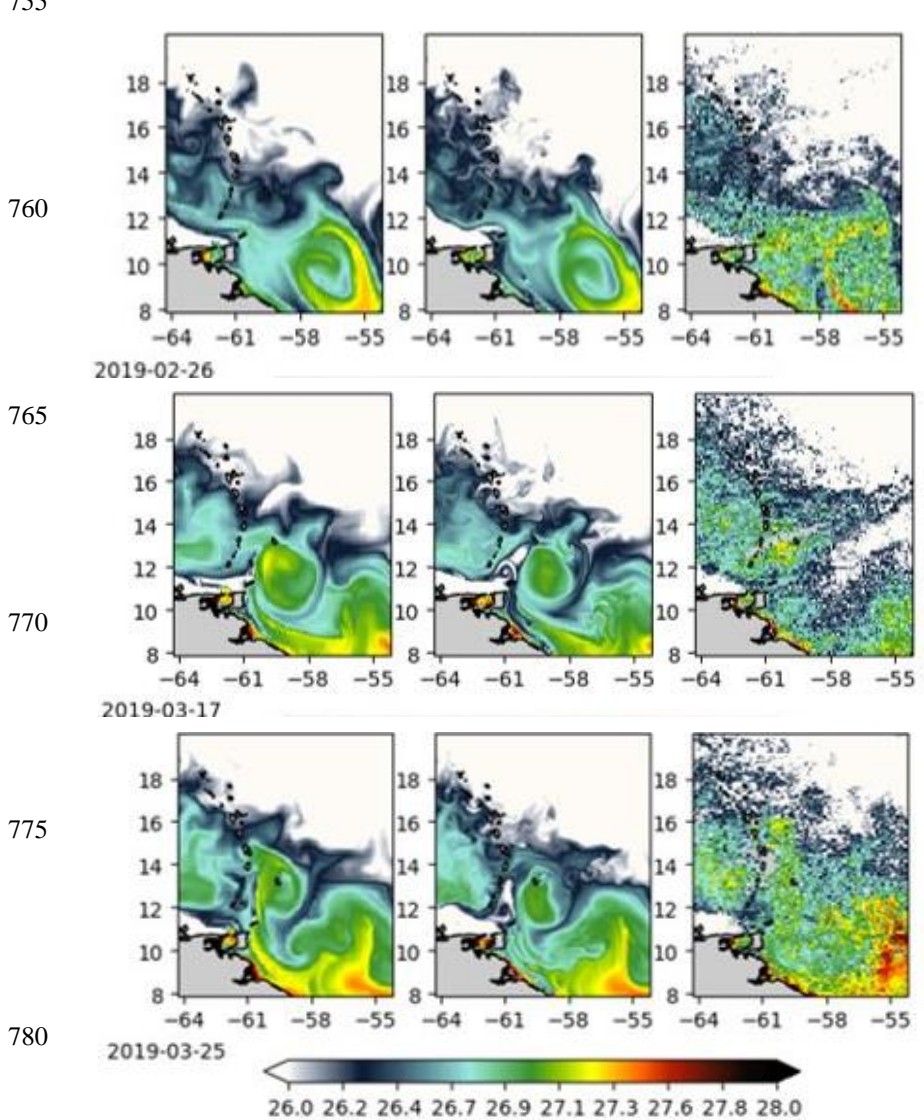

**Figure 8: Evolving SST signature of an NBC vortex. Top panel: situation on 26 February 2019, middle panel: situation on 17 March 2019 and bottom panel: situation on 25 March 2019. The maps from left to right represent the SST for the GLO12 model, the CAR36 model and the observed L3S SST respectively.**



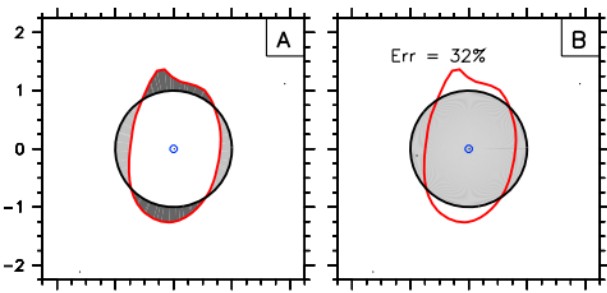

**Figure 9: Shape error from Kurian et al (2011). In red: the detected contour, in black the optimal circle closest to the contour. Here for illustration, a shape error of 32% is considered. Left panel, shaded: the difference between the area of the detected contour and the contour of the nearest optimal circle. Right panel, shaded: the area of the optimal circle.**




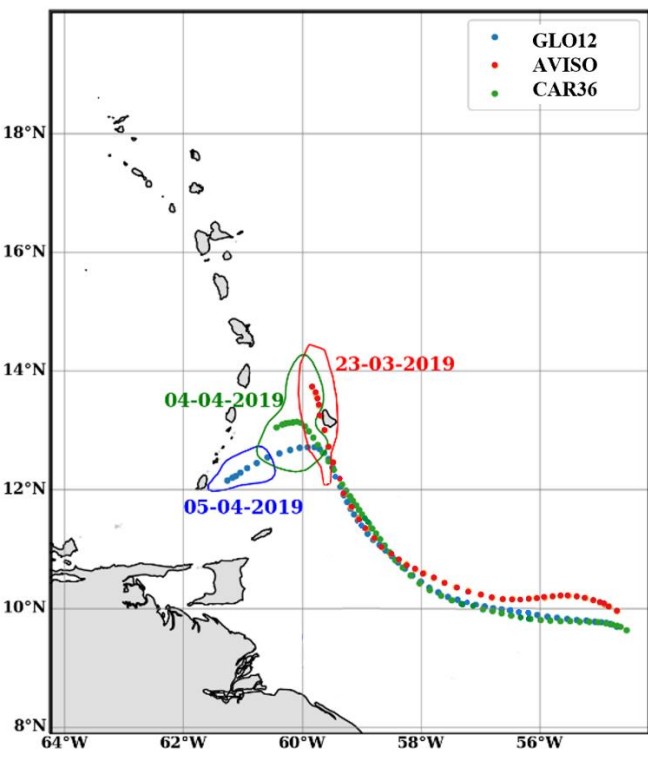

**Figure 10: Trajectories and shapes of an NBC eddy evolving from January to April 2019 in the Caribbean arc region, based on observed AVISO SSH data (red), simulated by GLO12 (blue) and simulated by CAR36 (green). For each eddy shape shown, the date is indicated.**

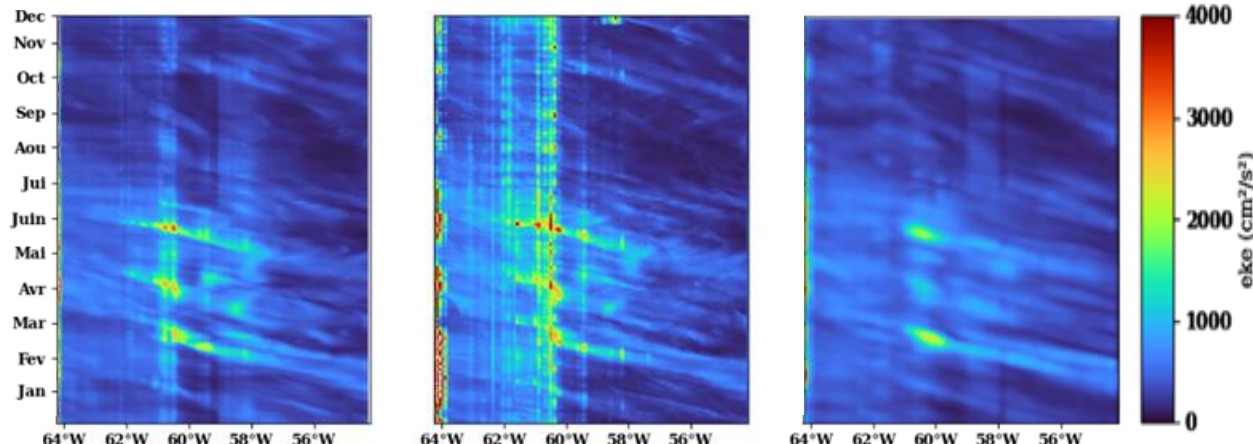

**Figure 11: Hovmöller diagram (longitude/time) of turbulent kinetic energy in the Caribbean arc region, for the year 2019, calculated from total current velocities from GLO12 (left panel) and CAR36 (centre panel), and fom geostrophic current velocities deduced from L4 AVISO SSH (right panel).**






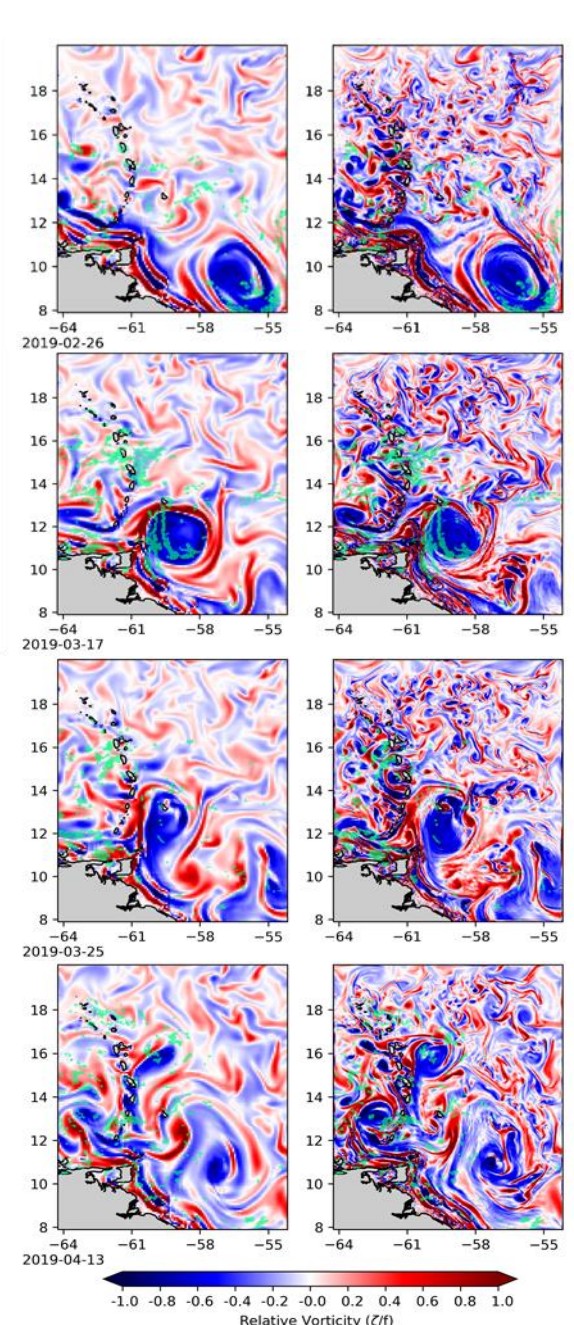



**Figure 12: Relative vorticity field normalised by planetary vorticity (Coriolis parameter f) and superposition of Sargassum detections (green pixels). The left panels correspond to the vorticity field obtained from GLO12 and the right panels to the vorticity field obtained from CAR36. From top to bottom, the fields for the dates 26 February, 17 March, 25 March and 13 April 2019 are shown respectively.**


