# Peer review of "CAR36, a regional high-resolution ocean forecasting system for improving drift and beaching of Sargassum in the Caribbean Archipelago."

_Geoscientific Model Development, 2023_

## Author Comment (AC3)

**Answer to the referee #1**

**Review Ms GMD-2023-183 ''CAR36, a regional high-resolution ocean forecasting system for improving drift and beaching of Sargassum in the Caribbean Archipelago' by Sylvain Cailleau et al.**

**General referee comments**

*The Ms present a new solution of IBI system based on NEMO ocean model at 1/36º resolution in the Caribbean Archipelago in order to forecast the drifting of Sargassum in the area. The system is forced at the open boundaries by daily solutions of the GLO12 and forced hourly by ECMWF.*

*The Ms. present an interesting application of a high resolution model with undoubtedly interest for statistical studies of submesoscale processes in the Caribbean area. The ability of the system to reproduce the local dynamics is tested qualitatively and quantitatively against SSH, SST, sargassum detection, drifting buoys and eddy tracking. The different metrics presented and the comparison with data are in line with the usual techniques used to test model agreement **although some effort in line with lagrangian diagnosis would be desirable**.*

*The paper is well written and provides a good introduction on the problem of drifting objects specifically in the area of interest. **In my opinion testing the model with an eddy in the area, although interesting, is far from the objective of the system which as the authors state is the drifting and 'beaching' of Sargassum.** Besides, **the effects of waves, very important in the east side of the domain, have not been considered.***

*I think the Ms can be published after some clarifications.*

**Authors' answer**
Dear Referee,

We would like to thank the referee for their constructive reviews and comments of the manuscript. We have addressed each comment individually in this document and in the revised manuscript, and provided explanations where needed. We believe this revised version fulfills the requirements of the journal by clarifying specific aspects and by improving the overall clarity of the paper .

**Main referee comments**

*Although being a research area that is very relevant, the pres*ent manuscript has two drawbacks.

**Main referee comment #1**
*First, the effect of Stokes drift in the drifting of objects is never treated nor discussed. How does this component of the velocity influence the lagrangian dynamics in the area?*

**Authors' answer**

We agree with the reviewer that indeed the wave impact is not mentioned in the ms. The reasons are set out below.

It is unclear to what extent Stokes drift affects lagrangian transport in the area. *Dagestad et al. (2019)* show the impact of the use of the Stokes drift in the drift calculation could be significant for fully submerged objects but did not improve the outcomes regarding objects exposed to surface wind, such as the floating Sargassum rafts.

Stokes drift component was not taken into account in the calculation of the CAR36 ocean surface current as CAR36 is primarily dedicated to the Sargassum drift forecasting by MétéoFrance, who only require for this purpose CAR36 sub-surface current (100m averaged current and current under the Ekman layer) as ocean current forcing for their drift model, called MOTHY. As the Stokes drift is a surface component of the ocean current, it is negligible in the sub-surface layers. From the sub-surface current forcing, MOTHY can get the mean currents, the regional and coastal currents, the mesoscale eddies and the sub-mesoscales structures. MOTHY does not use the Stokes drift either, only the wind is taken into account (*Daniel, 1996*). Several studies at MétéoFrance show no concluding impact of the Stokes drift in MOTHY.

Moreover, no wave coupling nor forcing has been applied to CAR36 model. The only wave effect appears in the definition of the drag coefficient when computing the wind stress through the bulk formulae which takes into account the sea surface roughness by using the Charnock coefficient (a constant non-dimensional surface roughness). Thus the wave induced air-sea momentum flux is partially considered in the CAR36 model configuration and consequently a part of the wave induced ocean vertical mixing as well. These wave effects on the wind stress must be taken into consideration especially on the Caribbean Archipelago area in the path of the tropical cyclones *(Moon et al, 2004)*.

For all these reasons, it was decided to not consider Stokes drift in the CAR36 setup for this application. We could consider a total ocean surface current by adding to CAR36 current a Stokes drift component from a wave model as it has been performed to generate the Copernicus Marine SMOC product (Surface and Merged Ocean Currents: GLOBAL_ANALYSIS_FORECAST_PHY_001_024-TDS ) but the validation of the drift forecasts must be carried out by MétéoFrance with MOTHY forced by the CAR36 sub-surface current. Moreover, we can note so far, following the study of *van Sebille et al. (2020)*, a total modelled current including the Stokes drift like SMOC (which take into account both the component of current from an ocean circulation model and the Stokes drift from a waves' model) didn't capture the general pathway of the undrogued Stokes drifters in the tropical Atlantic ocean.

Before providing CAR36 sub-surface current to MétéoFrance we wanted to fully assess our new system in a one year period of reference, firstly by using classical "macroscopic" validation metrics and secondly by using innovative diagnostics with a focus on the evolution of a particular NBC eddy until its dissipation on the Caribbean Archipelago, as it has been presented in ms. We've simply made the choice to validate model surface ocean parameters since most of observation data are available at the surface. We can suppose that the benefit of the CAR36 high resolution surface solution regarding the lower resolution GLO12 one, can be extrapolated to the subsurface current.

**Main referee comment #2**

*The second question is related to the objective of the paper. Comparing the performance of the model with the detection of Sargassum in the Eulerian framework is fine but, why not using a lagrangian metric (for instance a better way to test the difference between both solutions would be the use of the lagrangian divergence or FSLE). This second issue would give the ms. a more robust significance.*

As we explained previously, CAR36 has been developed to answer to a MétéoFrance need about Sargassum drift forecasting issue. CAR36 high resolution sub-surface eulerian current is dedicated to forcing MOTHY MétéoFrance operational lagrangian drift model to improve its forecasts against GLO12 forcing. The lagrangian analysis is not the objective of the present study. The aim of this ms consists in proving the benefit of having a higher horizontal resolution (CAR36/GLO12) for a best ocean current forcing of MOTHY and then an improvement of Sargassum drift forecasting. But we think the paper objectives are quite well explained in the ms. Consequently we focused on the eulerian ocean circulation since the lagrangian aspect will be assessed by MétéoFrance from its MOTHY simulations. First preliminary Lagrangian scores from MOTHY forced by CAR36 and GLO12 sub-surface currents have already shown an improvement of MOTHY drift forecasts. These results (summarized in the table below) have been performed by *Lina Pitek et al, 2023* for its engineering school internship at MétéoFrance (supervised by *Pierre Daniel*). Several observed trajectories of Sargassum drafts in 2019 (during a period of 24 hours) deduced from Sargassum satellite detection data, are compared to the trajectories obtained by MOTHY drift forecasts by considering GLO12 and CAR36 sub-surface currents' forcings. The table shows the mean, the median and the standard deviation of the distances between the observed and simulated trajectories after 24 hours decrease by using CAR36 forcing.

| | GLO12 Forcing | | CAR36 Forcing | |
|---|---|---|---|---|
| | Eckman | 100m | Eckman | 100m |
| **Mean(distances)** | 9.69 | 13.79 | 9.14 | 10.06 |
| **Median(distances)** | 6.98 | 11.65 | 8.06 | 9.08 |
| **STD(distances)** | 7.9 | 9.5 | 6.9 | 4.4 |

*Table 1: Mean, median and standard deviation of the distances between the Sargassum rafts' trajectories and MOTHY drift forecast trajectories after 24 hours. GLO12 and CAR36 currents under the Ekman layer and also averaged in the 100 first meters are applied as forcing to MOTHY. The year 2019 is considered.*

Even if proper lagrangian diagnostics such as FSLE analysis didn't presented in the ms., we compared nevertheless (fig. 6 in the ms) the lagrangian trajectory of the drifters and the eulerian current induced trajectories of CAR36 and GLO12. Besides we also performed a seeding experiment (from the lagrangian OceanParcels tool: https://oceanparcels.org/) by considering lagrangian particles inside a NBC eddy and their evolution in order to highlight the different solutions between CAR36 and GLO12 especially when the eddy arrived to the Archipelago. And in the ms it has been shown the considered NBC eddy evolution scenario obtained by CAR36 was more realistic especially in the eddy's end of life. We agree that a lagrangian FSLE analysis could provide further informations, in particular with regards to the way convergence and divergence zones are represented, but the lack of observation data to confront obtained results precludes its use as a validation tool. As such, we did not plan to develop this kind of diagnostics for this study.

**Specific referee comments**

**Specific comment #1**
*Page 2 Ln 68. Define Caribbean Archipelago.*

**Authors' answer**
An additional description of the area has been added in this section of the ms:

"… These ocean currents will replace GLO12 ones in the Météo-France drift forecast system. Thus a best representation of the local ocean dynamics is expected on the CAR36 domain focused on the Caribbean Archipelago composed by Trinidad and Tobago, Barbados islands and Lesser Antilles and which extends from 64.25°W to 54.17°W in longitude and from 7.89°N to 20.08°N in latitude."

**Specific comment #2**
*Section 2 (last paragraph). Why are you concerned specifically about eddies?*

**Authors' answer**
As it has been explained in the last paragraph of the section 2, the representation of the evolution and dissipation of an eddy (ie. by considering mesoscale and submesoscale structures) can prove the benefit of the HR CAR36 solution vs lower resolution GLO12 one. The latter cannot properly resolve such fine structures, especially when the eddy reaches the archipelago and start to dissipate. The more accurate will be the current to represent such fine structures, the more accurate will be the Sargassum drift forecasts deduced from the MOTHY MétéoFrance model forced by this current.

**Specific comment #3**
*Section 3.2. Ln 157 . Although this is true for the CAribbean and the west part of the domain this is not true for the east side.*

**Authors' answer**
We agree, even if tide is weak around the Archipelago, the west part of the area can be more impacted by the tide and then the tidal forcing in CAR36 setup is quite justified.

**Specific comment #4**
*Section 5.2. Ln 310. Is it possible to perform it?*

**Authors' answer**
It has been perfomed in the rest of the ms. The seeding experiment was allowed to highlight the difference in eddy evolution for CAR36 and GLO12. And the more qualitative following diagnostics allowed to conclude the end of life and dissipation of the eddy is more realistic in the case of CAR36.

**Specific comment #5**
*Section 5.2. Ln 319. It is very difficult to see from the L3 what the authors state (figure 8).*

**Authors' answer**
Maybe it will be clearer if we add red circle around the eddy SST signature:

[Figure]

2019-02-26

2019-03-17

2019-03-25

26.0 26.2 26.4 26.7 26.9 27.1 27.3 27.6 27.8 28.0

**Specific comment #6**

*Ln 353. <Can the authors discuss the disagreement respect the difference in the overestimation. Do you expect to improve the results using more eddies? (statistical relevance).*

**Authors' answer**

We consider in this study a one year calibration run of CAR36 and a unique case of NBC eddy. The large scale ocean circulation (up to the mesoscale) is constrained by the GLO12 solution through the spectral nudging method. And for the smaller scales, CAR36 is free. As a result, before reaching the archipelago, the representation of the NBC eddy as well as its azimuthal speed are similar and closed to the observation. But when the eddy arrived to the archipelago, its shape change and its diameter decrease for both models which are no more constrained by data assimilation (for GLO12) and spectral nudging (for CAR36). Without any constrain, the free model solutions can diverge from the observations and could explain the disagreement between the modeled eddy lifetime and the observed eddy one, and between the two models as well. But indeed a more statistical analysis from a longer simulation period

(several decades typically) would make the results more relevant. Now this longer simulation isn't expected.

**Specific comment #7**

*Figure 4. I suggest changing the scale for the bias and RMSE to remark on the differences.*

**Authors' answer**

The figure has been modified by changing the color palette scale, increasing thus the contrast. This new figure will replace the previous one in the ms.

[Figure]

**Specific comment #8**

*Figure 5 (the same comment).*

**Authors' answer**

Same modification.

[Figure]

2019-01-02 to 2019-12-24

**Additionnal bibliography references**

Daniel P., 1996: Operational forecasting of oil spill drift at METEO-FRANCE. Spill Science & Technology Bulletin . Vol. 3, No. 1/2, pp. 53-64.

Knut-Frode Dagestad, Johannes Rohrs, 2019: Prediction of ocean surface trajectories using satellite derived vs. modeled ocean currents. Remote Sensing of Environment 223, 130-142.

Pitek Lena, Pierre Daniel, Jean-Paul Boy, 2023: Prévoir la dérive des sargasses avec MOTHY. Rapport de stage pour l'obtention du diplôme d'Ingénieur de l'Ecole et Observatoire des Sciences de la Terre. http://www.meteorologie.eu.org/mothy/rapports/rapport_lena_pitek.pdf

Il-Ju Moon, Isaac Ginis, Tetsu Hara (2004): Effect of surface waves on Charnock coefficient under tropical cyclones. Geophys. Research Letters, vol. 31, L20302. https://doi.org/10.1029/2004GL020988.

Erik van Sebille, Erik Zettler, Nicolas Wienders, Linda Amaral-Zettler, Shane Elipot, Rick Lumpkin (2020): Dispersion of Surface Drifters in the Tropical Atlantic. Front. Mar. Sci, Vol. 7 https://doi.org/10.3389/fmars.2020.607426 .